# An M protein coiled coil unfurls and exposes its hydrophobic core to capture LL-37

**Piotr Kolesinski[1], Kuei-Chen Wang[1], Yujiro Hirose[2], Victor Nizet[2], Partho Ghosh[1]\***

[1]Department of Chemistry & Biochemistry, University of California, San Diego, La Jolla, United States; [2]Division of Host-Microbe Systems and Therapeutics, Department of Pediatrics, University of California, San Diego, La Jolla, United States

**Abstract** Surface-associated, coiled-coil M proteins of *Streptococcus pyogenes* (Strep A) disable human immunity through interaction with select proteins. However, coiled coils lack features typical of protein–protein interaction sites, and it is therefore challenging to understand how M proteins achieve specific binding, for example, with the human antimicrobial peptide LL-37, leading to its neutralization. The crystal structure of a complex of LL-37 with M87 protein, an antigenic M protein variant from a strain that is an emerging threat, revealed a novel interaction mode. The M87 coiled coil unfurled and asymmetrically exposed its hydrophobic core to capture LL-37. A single LL-37 molecule was bound by M87 in the crystal, but in solution additional LL-37 molecules were recruited, consistent with a 'protein trap' neutralization mechanism. The interaction mode visualized crystallographically was verified to contribute significantly to LL-37 resistance in an M87 Strep A strain and was identified to be conserved in a number of other M protein types that are prevalent in human populations. Our results provide specific detail for therapeutic inhibition of LL-37 neutralization by M proteins.

**\*For correspondence:**
pghosh@ucsd.edu

## Editor's evaluation

In this exciting article, the authors solve the crystal structure of a complex of LL-37 with the M protein M87. In this structure, the M87 coiled coil unfurled to exposed its hydrophobic core to interact with LL-37. These studies have provided important new information regarding the mechanism of interaction between coiled-coil proteins and the α-helical LL-37.

## Introduction

M proteins are the major surface-localized virulence factor of the widespread and potentially deadly bacterial pathogen *Streptococcus pyogenes* (Group A *Streptococcus* or Strep A) (***Ghosh, 2018***). One of the primary functions of M proteins is to enable Strep A to evade human innate and adaptive immune responses. This is brought about by interaction of M proteins with select human proteins. M proteins are antigenically sequence variable, with over 220 different types having been identified (***McMillan et al., 2013***). Despite this variation, the primary sequences of M proteins generally have a propensity to form dimeric, α-helical coiled coils, as verified by direct experimental evidence (***McNamara et al., 2008***; ***Macheboeuf et al., 2011***; ***Buffalo et al., 2016***). This propensity is easily distinguishable by the presence of heptad repeats (***Manjula and Fischetti, 1980***), in which amino acids in the *a* and *d* positions of the heptad are usually small and hydrophobic, and form the hydrophobic core of the coiled coil. In contrast to the usually complex topography of globular proteins, demarcated by pockets and

**eLife digest** We share our environment with many different bacteria. Some are beneficial for our health, like gut bacteria, but others can cause severe disease if they infect and spread within the body's tissues. For example, the bacterium *Streptococcus pyogenes* can cause conditions ranging from skin infections to a rapidly spreading deep-tissue infection, giving it the nickname "flesh-eating bacterium".

To prevent infection, our bodies have developed defence mechanisms that target disease-causing bacteria. These include antimicrobial molecules, such as LL-37, which is a small protein produced on the skin. LL-37 kills bacteria by puncturing their cell membrane (the bacterial equivalent of our skin); in other words, it acts like a tiny chemical dart that 'pops' the bacterial cell.

However, some bacteria, including *S. pyogenes*, can disarm these defences. *S. pyogenes* captures LL-37 on its surface with so called M proteins, which prevent LL-37 from reaching and destroying the underlying membrane. However, it was unknown how exactly the two proteins interact, especially since LL-37 is a simple molecule that lacks the structural features that allow most proteins to bind to each other.

Kolesiński et al. set out to determine how the M protein can 'grab' LL-37. A technique called X-ray crystallography allowed them to visualise the molecules atom by atom and to examine the configuration of the M protein after it had captured LL-37. The M protein selected for these experiments (M87) came from a strain associated with particularly severe disease, considered to be an emerging health threat. The results showed that M87 uncurled itself, thereby exposing specific parts that normally remain hidden. This way, it could capture LL-37, like a hand opening to grab an object.

Kolesiński et al. have revealed a key molecular mechanism that enables a disease-causing bacterium to invade our immune defences. Identifying which regions of M87 are involved in capturing LL-37 may help design more effective therapies to combat *S. pyogenes* infections.

cavities that enable specific protein–protein interactions, the simple fibrillar structure of M protein coiled coils raises the question of how M proteins achieve specific binding with their human targets.

A particular challenge lies in understanding how M proteins specifically bind and thereby neutralize the human antimicrobial peptide LL-37. This is because LL-37 also has a simple topography: LL-37 consists of a short amphipathic α-helix. LL-37 is a member of the cathelicidin antimicrobial peptide family and constitutes a major host immune defense against Strep A, as well as other pathogens (*Nizet et al., 2001*; *LaRock et al., 2015*; *LaRock and Nizet, 2015*). The 37-amino acid peptide is proteolytically generated from the precursor protein hCAP-18, which is produced by neutrophils, macrophages, mast cells, and keratinocytes, along with other epithelial cell types (*Nizet et al., 2001*; *Zaiou et al., 2003*; *Wong et al., 2013*). Like other amphipathic α-helical antimicrobial peptides, LL-37 functions by inserting into and lysing bacterial plasma membranes (*Sancho-Vaello et al., 2017*; *Sancho-Vaello et al., 2020*; *Schneider et al., 2016*). Notably, M1 protein was shown to neutralize the antimicrobial activity of LL-37 by sequestering it in a 'protein trap' on the Strep A surface and away from its target of action, the bacterial membrane (*LaRock et al., 2015*, *Lauth et al., 2009*). Not only does M1 protein attached to bacterial surface have this neutralization capacity, but so too does M1 protein released in soluble form from the Strep A surface, as occurs during infection (*LaRock et al., 2015*; *Herwald et al., 2004*). M1 protein also binds the LL-37 precursor hCAP-18 and consequently prevents the proteolytic generation of LL-37 (*LaRock et al., 2015*).

To determine the mechanism of LL-37 binding and neutralization by M proteins, we pursued co-crystallization. While M1 protein proved recalcitrant to co-crystallization, several new M protein types that bind and neutralize LL-37 were identified. LL-37 was co-crystallized with M87 protein, which is an M type from a strain that is an emerging health threat (*Turner et al., 2019*). The structure revealed a remarkable and novel mode of interaction for a coiled coil, in which a portion of the M87 protein coiled coil unfurled and exposed its hydrophobic core for interaction with LL-37. The LL-37-binding motif visualized in M87 protein was identified in a number of other M protein types, many of which are prevalent in human populations. Experimental evidence showed that some of these other M types bound and neutralized LL-37 similarly to M87 protein. Our results provide specific detail for inhibiting the interaction of M proteins with LL-37.

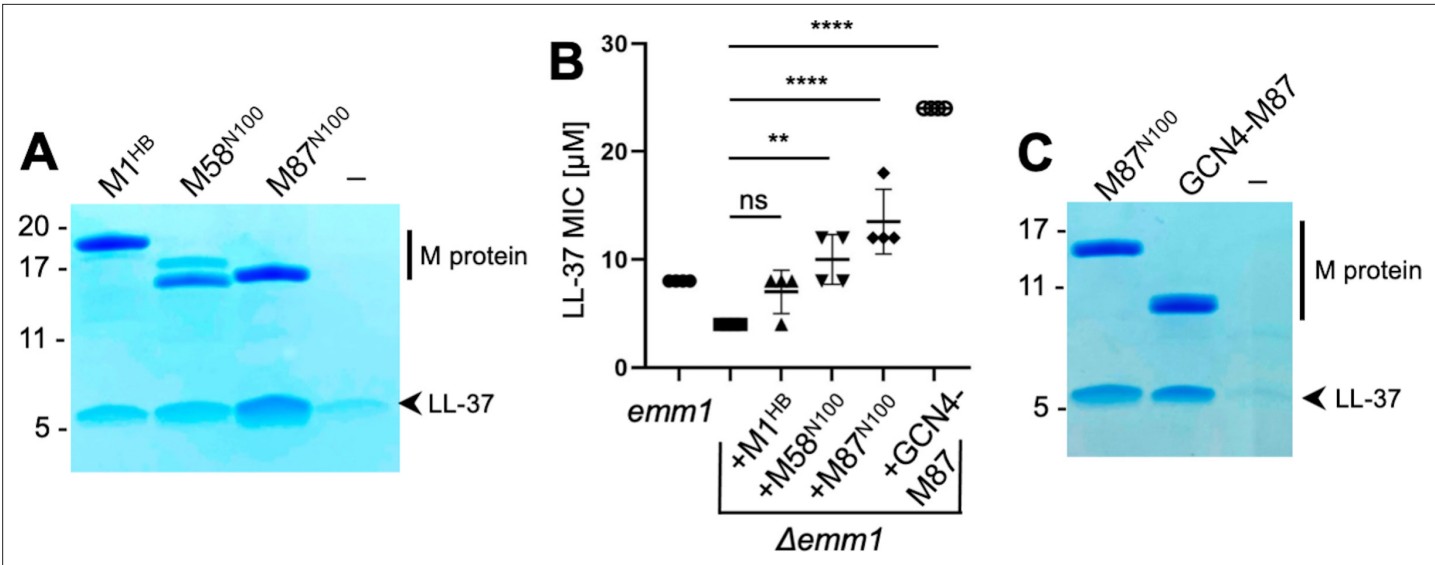

**Figure 1.** LL-37 binding and detoxification. (**A**) LL-37 binding to His-tagged M1[HB], M58[N100], or M87[N100] as determined by Ni[2+]-NTA agarose bead co-precipitation assay at 37°C. The last lane contains no M protein. Bound fractions were resolved by SDS-PAGE and visualized by Coomassie staining. The gel is representative of at least three experiments. *Figure 1—figure supplement 1* shows LL-37 binding to other M types. (**B**) LL-37 minimal inhibitory concentration (MIC) for wild-type Strep A *emm1* and isogenic *Δemm1* alone or supplemented with 10 μM M1[HB], M58[N100], M87[N100], or GCN4-M87. Data from four independent experiments are presented with means and standard deviations. Statistical significance was calculated using one-way ANOVA with Dunnett's post-hoc test. p-Values are as follows: ns > 0.05, *≤0.05, **≤0.01, ***≤0.001, and ****≤0.0001. (**C**) LL-37 binding to His-tagged M87[N100] or GCN4-M87, determined as in panel (**A**). The last lane contains no M protein. The gel is representative of at least three experiments.

The online version of this article includes the following figure supplement(s) for figure 1:

**Figure supplement 1.** LL-37 binding.

## Results

### Structure of the M87/LL-37 complex

To determine the mechanism of binding of LL-37 by M proteins, we pursued co-crystallization beginning with M1 protein but were unable to obtain co-crystals despite various attempts. A number of other M protein constructs (M4, M5, M22, M28, M44, M58, M77, M87, and M89), which were available in the laboratory for other purposes, were tried next. These consisted of the N-terminal 100 amino acids (denoted in the article with superscripted 'N100') of these M proteins. M58[N100] and M87[N100] bound LL-37, as did the M1[HB] fragment (*Figure 1A*), while the others did not bind LL-37 above background level (*Figure 1—figure supplement 1*). The level of LL-37 binding by M58[N100], M87[N100], and M1[HB] was approximately similar. As a point of comparison, the $K_D$ for the M1/LL-37 complex is ~1 μM (*LaRock and Nizet, 2015*).

To our knowledge, M58 and M87 proteins are the first M proteins besides M1 protein identified to bind LL-37. To determine whether this binding was functionally significant, M58[N100] or M87[N100] was exogenously added to an M1 Strep A strain in which *emm1* had been deleted (*Δemm1*) (*Lauth et al., 2009*). As previously shown (*LaRock et al., 2015*), *Δemm1* was sensitive to the antimicrobial action of LL-37 and exogenously added M1[HB] increased resistance against LL-37, although in this particular experiment the effect was not statistically significant (*Figure 1B*). Likewise, M58[N100] or M87[N100] exogenously added to *Δemm1* increased the LL-37 minimal inhibitory concentration (MIC), and in this case, these effects were statistically significant and M87[N100] had the greatest effect (*Figure 1B*).

Various fragments of M58 and M87 were tried in co-crystallization trials with LL-37. Success was had with a version of M87 protein (amino acids, aa 68–105) that had a portion of the canonical coiled-coil protein GCN4 (aa 250–278) (*LaRock et al., 2015*; *Lauth et al., 2009*; *O'Shea et al., 1991*) fused in register to its N-terminus. GCN4 fusion was pursued to stabilize coiled-coil formation in the M87 protein fragment, a technique that has proven successful in the crystallization of a number of coiled-coil proteins (*De et al., 2009*; *Li et al., 2002*). The structure of the GCN4-M87/LL-37 complex was determined through molecular replacement to 2.1 Å resolution limit (*Table 1*, *Figure 2*, *Figure 2—figure*

**Table 1.** Crystallographic data collection and model refinement.

| | GCN4-M87/LL-37 | GCN4-M87 |
|---|---|---|
| Data collection | | |
| Wavelength (Å) | 0.979 | 0.979 |
| Resolution range (Å) | 45.02 – 2.10 (2.18 – 2.10)* | 43.48 – 2.45 (2.54 – 2.45) |
| Space group | $P2_1$ | $P4_32_12$ |
| *Cell dimensions* | | |
| a, b, c (Å) | 50.2, 57.1, 71.5 | 97.2, 97.2, 40.7 |
| α, β, γ (°) | 90.0, 102.7, 90.0 | 90.0, 90.0, 90.0 |
| Total reflections | 134,344 (9365) | 91,495 (8197) |
| Unique reflections | 22,092 (1694) | 7580 (738) |
| Multiplicity | 6.1 (5.5) | 12.1 (11.1) |
| Completeness (%) | 94.51 (89.58) | 99.75 (100.00) |
| I/σ(I) | 8.50 (2.10) | 31.40 (0.78) |
| Wilson B-factor (Å$^2$) | 26.42 | 55.23 |
| $R_{meas}$† | 0.128 (0.735) | 0.404 (4.911) |
| $CC_{1/2}$‡ | 0.996 (0.878) | 0.987 (0.556) |
| Refinement | | |
| Resolution range (Å) | 45.02 – 2.10 (2.20 – 2.10) | 43.48 – 2.45 (2.80 – 2.45) |
| No. of reflections (work/test set) | 21,958/2426 | 7563/383 |
| $R_{work}/R_{free}$§ | 0.22/0.27 (0.27/0.34) | 0.26/0.27 (0.29/0.38) |
| *No. of non-hydrogen atoms* | 2673 | 1138 |
| Macromolecules | 2556 | 1126 |
| Ligands | 16 (ethylene glycol) | 2 ($PO_4^{3-}$) |
| Solvent | 101 | 2 |
| *r.m.s. deviations* | | |
| Bonds (Å) | 0.003 | 0.007 |
| Angles (°) | 0.46 | 0.95 |
| *Ramachandran plot* | | |
| Favored (%) | 100.00 | 100.00 |
| Outliers (%) | 0.00 | 0.00 |
| *Rotamer outliers (%)* | 0.00 | 0.00 |
| Clashscore | 3.37 | 3.98 |
| Average B-factor (Å$^2$) | 46.59 | 76.91 |
| Macromolecules | 46.80 | 75.60 |
| Ligands | 37.52 | 110.74 |
| Solvent | 42.81 | 64.27 |
| Number of TLS groups | 8 | 2 |
| PDB code | 7SAY | 7SAF |

*Table 1 continued on next page*

*Table 1 continued*

| | GCN4-M87/LL-37 | GCN4-M87 |
|---|---|---|
| Data collection | | |

Formulas for $R_{work}$ and $R_{free}$ are identical except 95% of the total number of reflections was used to calculate $R_{work}$, whereas the remaining 5% of reflection was used to calculate $R_{free}$.

*Values in parentheses are for the highest resolution shell.

$$R_{meas} = \sum hkl \sqrt{\frac{n}{n-1}} \sum_{j=1}^{n} |I_{hkl}\, j - \, < I_{hkl} > |\, /\sum hkl \sum j\, I_{hkl}\, j$$
†

‡$CC_{1/2}$ is the Pearson correlation between two half datasets.

§$R_{work} = \sum hkl\, |\, F_{obs} - F_{calc}|\, /\sum hkl F_{obs}$

*supplement 1A*). The GCN4-M87 fusion protein was verified to bind and neutralize LL-37, as gauged by co-precipitation and MIC assays, respectively (*Figure 1B and C*). GCN4-M87 bound LL-37 and led to a statistically significant increase in the LL-37 MIC, to an even greater extent than observed for M87[N100].

The structure revealed a single LL-37 molecule bound to a dimer of GCN4-M87 (*Figure 2A*). Two nearly identical 1:2 complexes occupied the asymmetric unit of the crystal (*Figure 2—figure supplement 1B*, RMSD 0.70 Å). Almost the entire length of LL-37, which was predominantly in α-helical conformation, was visible in the crystal structure. GCN4-M87 was likewise in α-helical conformation throughout, but strikingly, only the GCN4 portion formed a coiled coil while the M87 α-helices were unfurled and asymmetrically disposed. This was best appreciated by comparing the structures of the complexed and free form of GCN4-M87. The structure of the latter was determined to 2.4 Å resolution limit (*Table 1*, *Figure 2—figure supplement 1C*). Free GCN4-M87 formed a dimeric coiled coil throughout (*Figure 2B*), indicating that the unfurling of the M87 α-helices was unique to the bound form (*Figure 2C*). The greatest extent of unfurling occurred at Phe91, in which the distance between

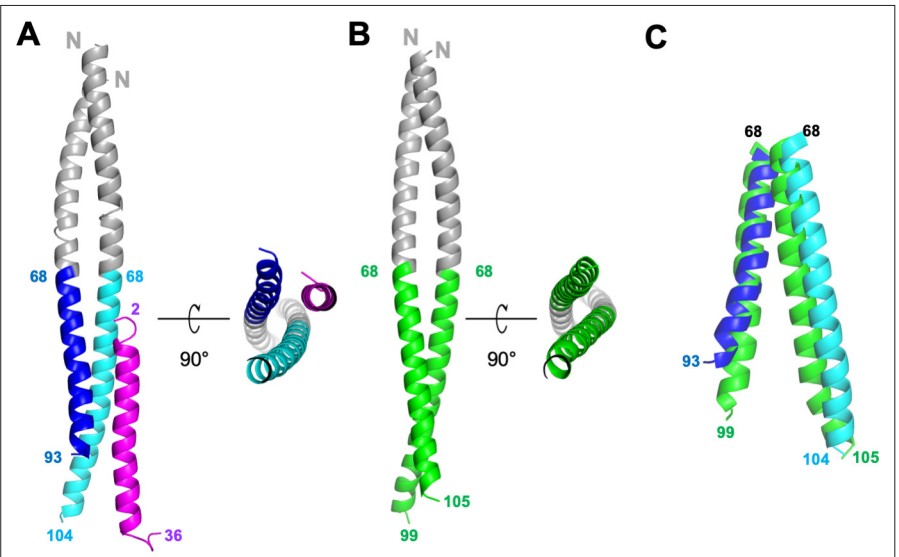

**Figure 2.** Structures of GCN4-M87/LL-37 complex and free GCN4-M87. (**A**) Cartoon representation of the GCN4-M87/LL-37 complex. LL-37 is in magenta. The GCN4 portion of GCN4-M87 is in gray, and the M87 portion in blue for the chain that makes more contacts to LL-37 (M87α1), and cyan for the chain that makes fewer contacts to LL-37 (M87α2). The N- and C-terminal amino acids for LL-37 and M87 are indicated. *Figure 2— figure supplement 1* shows electron density and superposition of the two GCN4-M87/LL-37 complexes in the asymmetric unit. (**B**) Cartoon representation of free GCN4-M87. The GCN4 portion is colored gray and the M87 portion green. (**C**) Superposition (based on GCN4, not depicted) of the M87 portions in bound (blue and cyan) and free (green) states.

The online version of this article includes the following figure supplement(s) for figure 2:

**Figure supplement 1.** Structures of the GCN4-M87/LL-37 complex and free GCN4-M87.



**Figure 3.** M87/LL-37 interface. LL-37 is in magenta, and the M87 α-helix that forms a greater number of contacts with LL-37 is in blue (M87α1) and the one that forms fewer contacts in cyan (M87α2). Shown are contacts formed with LL-37 (**A**) Phe5 and Phe6; (**B**) Ile 13; (**C**) Phe17; (**D**) Ile20, Val21, and Arg23; and (**E**) Ser9 and Lys12.

Cα atoms was 8.7 Å in the free form but 15.6 Å in the bound form. Only the M87 portion contacted LL-37, and therefore we will refer only to M87 rather than GCN4-M87 hereafter. The two M87 α-helices and the LL-37 α-helix together formed a parallel, three-helix bundle. One of the M87 α-helices made more contact with LL-37 (1016 Å² total buried surface area, average of the two complexes in the asymmetric unit of the crystal) (*Figure 3*, dark blue) while the other made less but still substantial

contact (491 Å$^2$ average total buried surface area) (**Figure 3**, cyan); to differentiate between the two M87 α-helices, the one making greater contact will be denoted M87α1 and the other M87α2.

## Interface dominated by hydrophobic interactions

Overall, hydrophobic interactions dominated the interaction site. Most notably, a number of hydrophobic M87 amino acids that occupied the core *a* or *d* positions and contacted each other in the free form, as is typical of coiled coils, were exposed in the bound form and instead contacted the hydrophobic face of the amphipathic LL-37 α-helix. Nearly five helical turns of LL-37 (Phe5-Val21) and M87 protein (Leu74-Trp92) engaged one another. Near the N-terminus of LL-37, a consecutive pair of phenylalanines, Phe5 and Phe6, was surrounded by hydrophobic *a* and *d* position amino acids of M87–Leu74 (*a*, usual position in heptad), Ala77 (*d*), and Tyr81 (*a*) from M87α along with Ala77 from M87α2 (**Figure 3A**). The two LL-37 Phe's and M87 Tyr81 formed a series of π-stacks. Two helical turns later in LL-37 was Ile13 that engaged in a ring of isoleucines, with M87 contributing Ile84 (*d*) from each of its helices (**Figure 3B**). A further helical turn later in LL-37 was Phe17 that packed against a pair of M87 leucines, Leu88 (*a*), one from each of the M87 helices, and π-stacked against M87α2 Phe91 (*d*) (**Figure 3C**). Lastly, one more helical turn further in LL-37 were Ile20, which was surrounded by M87α1 Phe91 (*d*) and Trp92; Val21, which packed against M87α2 Phe91; and Arg23 whose aliphatic side-chain atoms packed against M87α1 Trp92 (**Figure 3D**). These hydrophobic contacts were supplemented by a sparingly few polar ones, which occurred all within a small polar break in the hydrophobic face of LL-37, near its N-terminus: LL-37 Ser9 and Lys12 formed a hydrogen bond and salt bridge, respectively, with M87α1 Glu85 (**Figure 3E**), and in one of the two complexes, LL-37 Lys12 also formed a salt bridge with M87α1 Glu89.

## Importance of hydrophobic interactions

The role of the contacts observed structurally was evaluated through site-directed mutagenesis of M87 protein. To ensure that our structural observations were not limited to a fragment of M87 protein, these experiments were carried out with intact M87 protein. Intact His-tagged M87 protein and LL-37 were incubated at 37°C, and their interaction was determined through a Ni$^{2+}$-NTA agarose bead co-precipitation assay. Dual alanine substitutions of M87 Tyr81 (*a*) and Ile84 (*d*), which interacted with LL-37 Phe5, Phe6, and Ile13 (**Figure 3A and B**), significantly decreased LL-37 interaction (**Figure 4A and B**). M87 Y81A/I84A was verified through circular dichroism (CD) to have a similar structure to wild-type M87 protein at 37°C (**Figure 4—figure supplement 1A**). Additionally, M87 Y81A/I84A had a temperature-induced unfolding profile similar to that of wild-type M87 protein (**Figure 4—figure supplement 1B**), also as monitored by CD. These results indicated that alanine substitutions of Tyr81 and Ile84 affected LL-37 binding directly rather than indirectly through compromised structure, stability, or both. Alanine substitution of M87 Leu88 (*a*) and Phe91 (*d*), which interacted hydrophobically with LL-37 Phe17, Ile20, and Val21 (**Figure 3C and D**), likewise markedly decreased LL-37 interaction (**Figure 4A and B**). The secondary structure and stability of M87 L88A/F91A resembled that of wild-type M87 protein as well (**Figure 4—figure supplement 1A and B**). A substantial amount of the surface area of M87 Trp92 was buried by contact with LL-37, but surprisingly, substitution of this amino acid with alanine increased LL-37 interaction (**Figure 3C and D**). M87 Trp92 was adjacent to LL-37 Arg23 (**Figure 3D**), and thus we asked whether Arg substitution of M87 Trp92 would interfere with LL-37 binding. Indeed, M87 W92R was almost entirely deficient in LL-37 interaction (**Figure 4A and B**), while showing no changes in secondary structure or stability (**Figure 4—figure supplement 1A and B**).

The only M87 amino acid seen to make polar contacts in both complexes in the asymmetric unit of the crystal was Glu85 (**Figure 3E**). Ala substitution of M87 Glu85 had little effect on LL-37 binding (**Figure 4A and B**), suggesting that it did not contribute to affinity. To ask whether it instead contributed to specificity, we asked whether Arg substitution of this amino acid would decrease interaction with LL-37 as M87 Glu85 was adjacent to LL-37 Lys12. Consistent with our structural observations and a role in specificity, M87 E85R had significantly decreased interaction with LL-37 (**Figure 4A and B**). This decrease was a direct effect as M87 E85R had greater α-helical content than wild-type M87 protein and similar stability (**Figure 4—figure supplement 1A and B**).

These results validated the structural observations regarding the mode of interaction between M87 protein and LL-37, and indicated the importance of the M87 *a* and *d* heptad position amino acids to

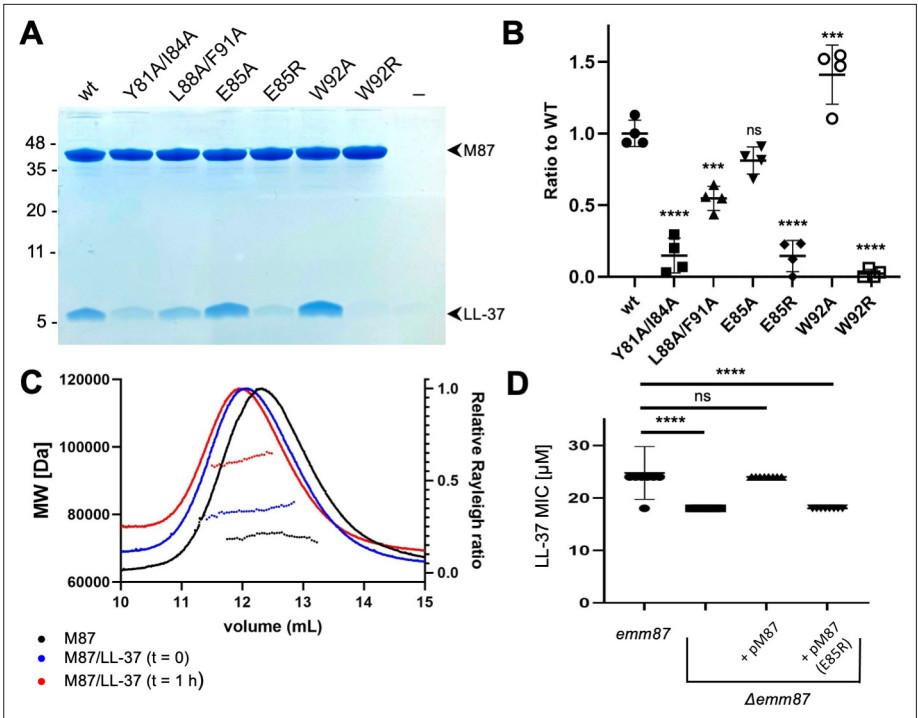

**Figure 4.** Evaluation of M87/LL-37 interactions. (**A**) LL-37 binding by His-tagged, intact wild-type M87 or M87 Y81A/I84A, L88A/F91A, E85A, E85R, W92A, or W92R at 37°C as determined by Ni$^{2+}$-NTA agarose bead co-precipitation. The last lane contains no M protein. Bound fractions were resolved by SDS-PAGE and visualized by Coomassie staining. The gel is representative of four experiments. *Figure 4—figure supplement 1* shows circular dichroism (CD) spectra and melting curves of wild-type and mutant M87 proteins. (**B**) Quantification of LL-37 binding to intact wild-type and mutant M87 proteins. The ratio between LL-37 and wild-type M87 protein band intensities was quantified in four independent experiments and used to determine a mean LL-37/wt-M87 ratio. Points shown are ratios between LL-37 and (wild-type or mutant) M87 protein band intensities, normalized by the mean LL-37/wt-M87 ratio. For each sample, means and SD are shown. Statistical significance was calculated using one-way ANOVA with Dunnett's post-hoc test. p-Values are as follows: ns > 0.05, *≤0.05, **≤0.01, ***≤0.001, and ****≤0.0001. (**C**) Molecular weight determination of intact M87 alone (black), M87 and LL-37 added together and immediately analyzed (blue), or M87 and LL-37 incubated for 1 hr before being analyzed (red) by size-exclusion chromatography (SEC) coupled to multiangle light scattering (MALS). Dotted lines indicate calculated molecular weights across the profile. Data are representative of three experiments. *Figure 4—figure supplement 2* shows SEC-MALS analysis of M87 protein and LL-37 incubated for 1 or 4 hr. (**D**) LL-37 minimal inhibitory concentration (MIC) for *emm87* and *Δemm87* alone or expressing wild-type M87 (pM87) or M87 E85R (pM87(E85R)) from a plasmid. Data from eight independent experiments are presented as mean ± SD. The statistical test and p-values are the same as described for panel (**B**).

The online version of this article includes the following figure supplement(s) for figure 4:

**Figure supplement 1.** Structure and stability of M87 mutant proteins.

**Figure supplement 2.** Self-limiting growth of the M87/LL-37 complex.

---

the interaction. They further indicated that the polar contact conferred by M87 Glu85 conferred specificity rather than binding affinity.

## Stoichiometry of interactions

For an LL-37 neutralization mechanism that involves an M protein trap (*LaRock et al., 2015*), the 1:2 LL-37:M87 stoichiometry was puzzling. LL-37 forms variably sized oligomers in solution (*Johansson et al., 1998*; *Oren et al., 1999*; *Xhindoli et al., 2016*), and thus we undertook solution phase studies of the complex through size-exclusion chromatography (SEC) coupled to multiangle light scattering (MALS). The molecular weight of intact M87 protein alone was 73.6±1.2 kDa (calc. 72.4 kDa) (*Figure 4C*, black). A 10-fold excess of LL-37 to the intact M87 dimer was added, and the sample was applied to SEC-MALS almost immediately after mixing. The complex had a mass of 80.4±0.6 kDa,

which corresponded to an M87 protein dimer bound to one or two molecules of LL-37 (calc. 4.5 kDa) (*Figure 4C*, blue). Notably, after 1 hr of incubation of LL-37 with M87 protein, the mass of the complex was 97.1±0.6 kDa (*Figure 4C*, red), which corresponded to an M87 protein dimer bound to five or six molecules of LL-37. Incubation of up to 4 hr also resulted in the same increased mass, indicating self-limiting growth of the complex (*Figure 4—figure supplement 2*). These results suggested that the single LL-37 molecule bound to an M87 dimer could recruit four or five additional LL-37 molecules.

### *emm87* confers resistance to LL-37

We asked whether the mode of interaction visualized through crystallography was applicable to M87 protein in its native conformation on the Strep A surface. An isogenic Δ*emm87* strain was constructed and complemented with a plasmid expressing either wild-type *emm87* or *emm87* containing the E85R substitution, the latter having greatly diminished LL-37 binding in solution (*Figure 4A*). The wild-type M87 strain was resistant to LL-37 (*Figure 4D*) and indeed had a greater LL-37 MIC than the M1 strain (*Figure 1C*). Deletion of *emm87* led to significantly increased sensitivity to LL-37 (*Figure 4D*). Notably, the Δ*emm87* strain complemented with wild-type M87 was restored to the LL-37 resistance of the wild-type parental M87 strain, while complemented with M87 E85R remained sensitive to LL-37, similar to the level seen for the uncomplemented Δ*emm87* strain. These LL-37 susceptibility results provide physiological validation for our structural observations.

### Conservation of M87 motif in other M types

The LL-37-binding motif visualized in M87 protein was identified in the sequence variable N-terminal regions of 14 other M protein types (*Figure 5A*). The motif consisted of two consecutive ideal heptads, that is, with *a* and *d* positions occupied by canonical hydrophobic amino acids (*Wagschal et al., 1999*; *Tripet et al., 2000*; *Kwok and Hodges, 2004*), including a strictly conserved Tyr at the *a* position of the first heptad. These hydrophobic amino acids in M87 protein (Tyr81, Ile84, Leu88, and Phe91) were shown above to be crucial to interaction with LL-37 (*Figure 4A and B*). Along with these, the motif had a hydrogen bond acceptor at the *e* position of the first heptad, which in M87 protein (Glu85) provided specificity by contacting LL-37 Lys12 (*Figure 3*). In addition, a positively charged amino acid was excluded from the *e* position of the second heptad, which in M87 protein was Trp92 and proximal to LL-37 Arg23 (*Figure 3*). Preceding these two ideal heptads was a nearly ideal heptad, with the *a* position occupied by canonical amino acids (i.e., Leu or Tyr) while the *d* position was tolerant of less than ideal amino acids (i.e., Ala77 as in M87, and also Gln) (*Tripet et al., 2000*; *Kwok and Hodges, 2004*).

Among the M proteins identified to have this motif were M25, M58, and M68 proteins. We showed above that M58$^{N100}$ bound LL-37 (*Figure 1A*). Similar fragments consisting of the N-terminal 100 amino acids were constructed for M25 and M68. M68$^{N100}$ bound LL-37 as did M25$^{N100}$ (*Figure 5B*). To test whether the LL-37-binding mode observed for M87 protein was conserved in these M proteins, Arg substitutions of the equivalent of M87 E85 were constructed. As the E85R substitution had been evaluated only in intact M87 protein, it was introduced into the M87$^{N100}$ fragment, which resulted in significantly decreased LL-37 binding as well (*Figure 5B and D*). Arg substitution of the equivalent Glu in M25$^{N100}$ (E87) and M68$^{N100}$ (E79) also led to significantly decreased LL-37 binding (*Figure 5B and D*). While Arg substitution of M58$^{N100}$ (E97) decreased LL-37 binding, this effect was not statistically significant. Modeling suggested that an Arg at this position in M58 was capable of salt bridging with M58 D94, thereby attenuating its LL-37 repulsion. In the case of M25 and M58 proteins, we noticed that the amino acid preceding the strictly conserved Tyr was Gly, which is a helix-breaking amino acid. Therefore, we substituted the Gly in M25 and M58 proteins with Asp, which is the equivalent amino acid in M87 (D80), and found that M25$^{N100}$ G82D and M58$^{N100}$ G92D had significantly higher levels of LL-37 binding compared to their wild-type counterparts (*Figure 5C and D*).

These results are consistent with M25, M58, and M68 binding LL-37 in the same mode as observed for M87 protein and support the hypothesis that the M87 protein LL-37-binding motif is conserved in at least 14 other M proteins, which belong to the E2 or E3 cluster of M proteins (*Sanderson-Smith et al., 2014*).

**Figure 5.** Conservation of the M87LL-37-binding motif. (**A**) M87 amino acids that make hydrophobic contact to LL-37 are boxed in red and those that disrupt interaction with LL-37 through Arg substitution are in blue boxes, as are amino acids that are similarly positioned in other M proteins. The arrow indicates the position of M87 D80. (**B**) LL-37 interaction with His-tagged wild-type or mutant $M87^{N100}$, $M25^{N100}$, $M58^{N100}$, $M68^{N100}$ proteins determined by $Ni^{2+}$-NTA agarose bead co-precipitation at 37°C. Bound fractions were resolved by SDS-PAGE and visualized by Coomassie staining. Data are

*Figure 5 continued on next page*

Figure 5 continued

representative of six experiments. (**C**) LL-37 interaction with His-tagged M25$^{N100}$, M25$^{N10}$ G82D, M58$^{N100}$, and M58$^{N100}$ G92D, determined as in panel (**B**). Data are representative of six experiments. (**D**) Quantification of the effect of mutagenesis on LL-37 binding. The ratio between LL-37 and wild-type M protein (i.e., M25, M58, M68, or M87) band intensities was quantified in six independent experiments and used to determine a mean LL-37/wt-M ratio. Points shown are ratios between LL-37 and mutant M protein band intensities, normalized by the corresponding mean LL-37/wt-M ratio. For each sample, means and SD are shown. Statistical significance was calculated using Student's $t$-test to compare mutant and the corresponding wild-type protein. p-Values are as follows: ns > 0.05, *≤0.05, **≤0.01, ***≤0.001, and ****≤0.0001.

## Discussion

We sought to understand how M proteins achieve specific binding of LL-37, which is essential to neutralization of this human antimicrobial peptide by the M1 strain of Strep A and likely by other strains as well (*Nizet et al., 2001*; *LaRock et al., 2015*; *Lauth et al., 2009*). This question was challenging as the fibrillar structure of M proteins excludes features commonly present in globular proteins that enable specific binding, such as pockets or cavities. Adding to the challenge, the structure of LL-37 lacks complexity, consisting simply of an amphipathic α-helix. We discovered a remarkable mode of interaction of M87 protein with LL-37 through structure determination. The two α-helices of M87 protein in the free state formed a coiled coil, with amino acids in the core *a* and *d* positions engaged in 'knobs-into-holes' packing. In the bound state, the two helices did not form a coiled coil, which was instead unfurled with the helices asymmetrically disposed. Intriguingly, analysis of 'knobs-into-holes' packing of M87 in the free state shows that A77 and Y81 are not ideally packed (*Kumar and Woolfson, 2021*). This suggests that the segment containing A77 and Y81 may initiate unfurling to bind LL-37. Most significantly, in the bound state, hydrophobic amino acids at the M87 *a* and *d* positions were exposed and formed a continuous patch that contacted the hydrophobic face of the LL-37 α-helix. These hydrophobic contacts dominated the interaction interface and were shown to be essential by Ala substitution mutagenesis. Recently, a coiled-coil dimer that binds an α-helical partner in a roughly similar manner was reported (*Dimitrova-Paternoga et al., 2021*), but in that complex coiled-coil character is maintained, unlike the loss of coiled-coil character observed for the M87/LL-37 complex. In addition, while coiled-coil asymmetry coupled to cognate partner binding has been noted in some proteins (*Sato et al., 2007*; *Noell et al., 2019*), this has taken the form of a helical stagger with coiled-coil character maintained. To our knowledge, the unfurling and asymmetric exposure of consecutive *a* and *d* position amino acids to form a continuous interaction site is a novel binding mechanism for a coiled-coil protein.

The LL-37-binding site in M87 protein was formed by two ideal heptads preceded by a nearly ideal heptad (Ala and Gln being tolerated at the *d* position). M proteins identified or predicted to bind LL-37 in the same mode also had at least three such consecutive repeats, with the first heptad being ideal in most cases (*Figure 6A*, *Figure 6—figure supplement 1A*). In contrast, M proteins identified not to bind LL-37 lacked consecutive ideal or near-ideal heptads. Instead, ideal heptads occurred as isolated singletons in the midst of nonideal heptads. Notably, M1 protein has only an isolated ideal repeat (*Figure 6A*), indicating that there are additional mechanisms for binding LL-37, different from that observed for M87 protein. In general, the occurrence of two or more consecutive ideal heptads is rare in M protein variable regions (*Figure 6—figure supplement 1*), whose sequences frequently have coiled-coil destabilizing amino acids at the *a* and *d* positions (e.g., Glu or Lys) (*McNamara et al., 2008*; *Nilson et al., 1995*). In the case of the M1 B-repeats region, nonideal heptads have been shown to be functionally essential for creating protein dynamics, which are required for binding fibrinogen in a 'capture-and-collapse' mechanism (*Stewart et al., 2016*). Dynamics in the consecutive ideal repeats of M87 and related M proteins are likely to be much lower in magnitude than in the nonideal M1 B-repeats, but nevertheless, it appears that enough breathing motion exists even in these ideal coiled coils for the infiltration of LL-37 and disruption of coiled-coil structure. The loose 'knobs-into-holes' packing of Y81 in the free form of M87, along with the conservation of the Tyr in M proteins shown or implicated to bind LL-37, suggests that this may be a general site for initiating unfurling.

The single molecule of LL-37 bound to M87 protein acted as a nucleator for the recruitment of additional LL-37 molecules. One or two molecules of LL-37 molecule bound to M87 protein in solution at an initial time point, but grew to five or six molecules over time, as evidenced by SEC-MALS analysis. This is consistent with observations that LL-37 oligomerizes in solution (*Johansson et al., 1998*; *Oren et al., 1999*; *Xhindoli et al., 2016*). Plausible mechanisms for the recruitment of additional LL-37



**Figure 6.** LL-37-binding motif and higher-order complex formation. (**A**) Occurrence of ideal heptads (dark gray boxes; Ile, Leu, Met, Phe, Tyr, or Val at both *a* and *d* positions) and near-ideal heptads (light gray boxes; ideal amino acid at *a* position and Ala at *d* position, with M25 being an exception with Gln at this position) within the N-terminal regions of various M protein types. Within these boxes, yellow circles indicate Tyr at an *a* position; green circles Asp, Asn, Glu, or Gln at an *e* position; and red circles Arg or Lys at an *e* position. Yellow and red circles favor LL-37 binding, while the red circles disfavor LL-37 binding. *Figure 6—figure supplement 1* shows the occurrence of ideal and near-ideal heptads within the N-terminal regions of additional M types. (**B**) Model of higher-order M87/LL-37 complex. M87α1 and M87α2 are in blue and cyan, respectively, and LL-37 is in magenta. Left: LL-37 bound to M87 protein has its polar face free to form polar contacts with a second LL-37 molecule (light gray, as seen in a structure of LL-37 alone, PDB 7PDC). Right: the light gray LL-37 has its hydrophobic face free to form hydrophobic contacts with yet another LL-37 molecule (dark gray, as seen in another structure of LL-37 alone, PDB 5NNM). The light and dark gray LL-37 molecules do not contact M87 protein.

The online version of this article includes the following figure supplement(s) for figure 6:

**Figure supplement 1.** LL-37-binding motif.

molecules are suggested by crystal structures of LL-37 alone. In one crystal structure, the polar faces of two LL-37 molecules contact one another in antiparallel orientation (*Sancho-Vaello et al., 2020*), and in another, the hydrophobic faces do likewise (*Sancho-Vaello et al., 2017*). Thus, it is possible that the single LL-37 molecule bound through its hydrophobic face to M87 protein is able to recruit a second molecule of LL-37 through polar face–polar face interactions. This second LL-37 molecule would then be able to recruit a third molecule of LL-37 through hydrophobic face–hydrophobic face

interactions (*Figure 6B*). While this sort of growth is not self-limiting, we observed that the LL-37/M87 complex was self-limiting at five or six molecules of LL-37 per M87 dimer, and thus the specific details of LL-37/LL-37 interactions are likely to differ. The excess of LL-37 bound by M87 protein is consistent with a 'protein trap' neutralization mechanism (*LaRock et al., 2015*).

The physiological relevance of the mode of LL-37 binding by M87 protein was established through deletion and complementation experiments. A deletion of *emm87* resulted in significant sensitivity to LL-37, as seen in the decrease in the LL-37 MIC. Resistance against LL-37 was restored to the Δ*emm87* strain by a plasmid encoding wild-type *emm87* but not *emm*87 (E85R). In vitro studies using purified proteins showed that M87 E85R does not bind LL-37, and structural studies provided an explanation for this – M87 E85 is positioned next to LL-37 Lys12.

M87 Strep A strains are prevalent in human populations and are an emerging cause of human clinical disease (*Turner et al., 2019*; *Steer et al., 2009*; *Li et al., 2020*). The genomes of almost all M87 strain isolates contain a recombination event that increases the expression of Strep A toxin genes (NADase and streptolysin O) (*Turner et al., 2019*). Importantly, the same recombination event is present in the M1 strain that caused a global pandemic starting in the 1980s and an M89 strain that is responsible for an ongoing epidemic (*Zhu et al., 2015*). A few isolates of the M82 strain also carry this recombination event (*Turner et al., 2019*). The M82 strain, along with a number of the other M types identified to have the M87 motif, are prevalent in human populations, including M25, M68, M90, and M103 (*Turner et al., 2019*; *Steer et al., 2009*; *Li et al., 2020*). These observations provide motivation for pursuing therapeutic inhibition of LL-37 binding by M proteins, and our results provide specific detail to achieve this end.

## Materials and methods
### Bacterial strains and culture conditions
*S. pyogenes* strains M87 20161436 (NCBI SRA accession: SAMN07154152) and its isogenic Δ*emm87* strain (*Hirose et al., 2022*), and M1T1 5448 and its isogenic Δ*emm1* strain (*Lauth et al., 2009*) were used. *S. pyogenes* was grown as standing cultures in Todd-Hewitt broth in ambient atmosphere at 37°C. *Escherichia coli* was cultured in lysogeny broth at 37°C with agitation. For selection and maintenance of strains, antibiotics were added to the medium at the following concentrations: erythromycin 500 µg/ml for *E. coli* and 2 µg/ml for *S. pyogenes*; chloramphenicol, 2 µg/ml for *S. pyogenes*.

### DNA manipulation
Coding sequences for M proteins were cloned into a modified pET28b vector (Novagen) that contained sequences encoding an N-terminal His$_6$-tag followed by a PreScission protease cleavage site (*Buffalo et al., 2016*). Amino acid substitutions and deletions were introduced into pET28b vectors with the QuikChange II Site-Directed mutagenesis kit (StrataGene), according to the manufacturer's directions, and into pM87 E85R (*Hirose et al., 2022*), which was used for expression in *S. pyogenes*, with the Phusion Site Directed Mutagenesis Kit (Thermo Scientific, Waltham, MA). The coding sequence for GCN4 250–278 was subcloned from *Saccharomyces cerevisiae* and was fused to M87 68-105 through strand overlap extension PCR. An M87 protein-expressing vector (pM87) was constructed by insertion of *emm87* into pDCerm (*Hirose et al., 2022*).

### Peptides and proteins
LL-37, which was chemically synthesized and lyophilized as a fluoride salt (GenScript, 95% purity), was solubilized at 5 mg/ml in sterile deionized water for MIC assays or 100 mM NaCl, 20 mM HEPES-NaOH, pH 7.5 (HS) for other experiments.

Expression and purification of M proteins constructs were carried out as previously described (*McNamara et al., 2008*; *Buffalo et al., 2016*), except for the following minor modifications. After PreScission protease digestion, GCN4-M87 was subjected to gel filtration chromatography using a Superdex 200 (GE Healthcare) column equilibrated with HS buffer. For formation of the GCN4-M87/LL-37 complex, GCN4-M87 (2 mg/ml) was mixed with a threefold molar excess of LL-37 (3 mg/ml), both in HS buffer, and the complex was purified by gel filtration chromatography using a Superdex 200 column that had been equilibrated with 100 mM NaCl, 20 mM MES-NaOH, pH 6.5. Intact wild-type and mutant M87 proteins, following Ni$^{2+}$-NTA agarose bead purification, were subjected to gel

filtration chromatography on a Superdex 200 column that had been equilibrated with HS buffer. For CD measurements, the His-tag on intact wild-type and mutant M87 proteins was removed by PreScission protease digestion, and the cleaved product was further purified by reverse $Ni^{2+}$-NTA chromatography. Protein concentrations of M proteins were determined by measuring $A_{280}$ with the sample in 6 M guanidine hydrochloride, 20 mM Tris, pH 8.5, and using a calculated molar 280 nm extinction coefficient. The concentration of LL-37 and GCN4-M87/LL-37 complex was measured using the Bradford assay (Bio-Rad) with BSA as a standard.

## Co-precipitation assay

One and half nmol of $His_6$-tagged M protein constructs (2–10 µl) were added to 50 µl of $Ni^{2+}$-NTA agarose beads that had been pre-equilibrated with HS buffer and incubated with gentle agitation for 10 min at room temperature (RT). Beads were centrifuged (3000 × $g$, 30 s, RT) and the supernatant was removed. 6 nmol of LL-37 in 150 µl of HS buffer was added to the beads and incubated with gentle agitation for 30 min at 37°C. The beads were washed three times each with 1 ml HS buffer containing 5 mM imidazole, pH 8.0. For the washes, the resin was mixed with the wash solution by gentle agitation, incubated for 1 min at RT, and then centrifuged (3000 × $g$, 30 s, RT). Bound proteins were eluted with 50 µl HS containing 400 mM imidazole, pH 8.0. Protein samples were resolved by SDS-PAGE and stained with InstaBlue (APExBIO). The intensity of gel bands was quantified with ImageJ (*Schneider et al., 2012*) and confirmed to be within the linear range of detection. As the intensity of each gel depended on the length of time of staining and destaining, a normalization factor was used. For this, the ratio of the LL-37 band intensity to wild-type M protein band intensity from a particular gel lane was determined from three or more independent gels, and the mean of these ratios (mean of LL-37/$M_{WT}$) was used as the normalization factor. Accordingly, for each data point (i.e., gel lane), the ratio of the LL-37 band intensity to M protein (wild-type or mutant) band intensity was quantified and divided by the normalization factor (mean of LL-37/$M_{WT}$).

## Minimal inhibitory concentration

*S. pyogenes* that had been grown overnight were inoculated into Todd-Hewitt broth at 1:100 dilution and grown at 37°C to an $OD_{600}$ of 0.4. The culture was diluted to an $OD_{600}$ of 0.1, and 5 µl (~$10^5$ CFU) was mixed into 100 µl of RPMI 1640 medium with glutamine, which contained 0, 2, 4, 8, 12, 18, or 32 µM LL-37. In some experiments, the medium also contained 10 µM $M1^{HB}$, $M58^{N100}$, $M87^{N100}$, or GCN4-M87 protein. *S. pyogenes* were grown in individual wells of a 96-well plate for 24 hr at 37°C. *S. pyogenes* viability was assessed at this time point by the color of the RPMI medium, where yellow indicated bacterial growth and red no bacterial growth. The MIC was defined as the LL-37 concentration at which no growth was detectable at 24 hr.

## Molecular mass determination

Intact His-tagged M87 protein (2.5 mg/ml) alone or mixed with LL-37 (1.6 mg/ml; 10-fold molar excess over M87 dimer) in HS (100 mM NaCl, 20 mM HEPES-NaOH, pH 7.5) was centrifuged (10 min, 20,000 × $g$, 20°C) to remove aggregates. Samples (100 µl) were then either immediately applied to a Superdex 200 10/300 column that had been pre-equilibrated in HS or incubated 1–4 hr at RT before application to the column. Samples eluting from the column were monitored with a light scattering detector (DAWN HELEOS II, Wyatt Technology, Santa Barbara, CA) and a differential refractometer (Optilab T-rEX; Wyatt Technology). Data processing and molecular mass calculation were performed with ASTRA software (Wyatt Technology).

## Crystallization and data collection

Crystallization trials were carried out at 293 K using the hanging drop vapor diffusion method. The GCN4-M87/LL-37 complex was concentrated by ultrafiltration using a 3500 MWCO membrane (Millipore; 4500 × $g$, 30 min, 15°C) to 8 mg/ml. GCN4-M87 alone was concentrated to 10 mg/ml by ultrafiltration through 3500 MWCO membrane (Millipore; 4500 × $g$, 30 min, 4°C) The complex was brought to RT before introduction into crystallization drops to overcome its low solubility at 4°C.

The GCN4-M87/LL-37 complex (0.8 µl) was mixed in a 1:1 ratio with 5% (v/v) acetonitrile, 0.1 M MES-NaOH, pH 6.5. Microclusters of plates (ca. $20 \times 20 \times 5$ µm³) that had grown after 2–4 days were crushed, diluted 125-fold with the precipitant solution, and centrifuged (2000 × $g$, 30 s, RT).

The supernatant was collected and used as a seed stock. GCN4-M87/LL-37 (0.9 µl) was mixed with 0.9 µl of 10% (v/v) acetonitrile, 0.1 M MES-NaOH, pH 6.5, and 0.2 µl of the seed stock. Clusters of diffraction-sized crystals (50–150 µm in each dimension) that were obtained after 3–7 days were crushed once again and used for a subsequent round of seeding, carried out in the same manner as the first round. These two rounds of seeding yielded single crystals (200 × 200 × 10 µm³) that were cryo-preserved by three serial transfers to the precipitant solution supplemented with 10, 20, and 30% ethylene glycol, respectively, and flash-cooled in liquid $N_2$.

GCN4-M87 (0.8 µl) was mixed in a 1:1 ratio with 0.6 M monobasic ammonium phosphate, 0.1 M Tris-HCl, pH 8.0. Single crystals (300 × 150 × 150 µm³), which were obtained after 4–7 days, were cryo-protected in the precipitant solution supplemented with 30% ethylene glycol and flash-cooled in liquid $N_2$ prior to data collection.

Diffraction data were collected at SSRL (beamline 9–2) at 0.979 Å, integrated with Mosflm (GCN4-M87/LL-37) (*Battye et al., 2011*) or DIALS (GCN4-M87) (*Winter et al., 2018*) and scaled with Aimless (*Evans and Murshudov, 2013*; *Table 1*). Because of the highly anisotropic nature of both datasets, the resolution cutoffs for both were determined using anisotropic $CC_{1/2}$.

## Structure determination and refinement

Phases for the GCN4-M87/LL-37 complex and free GCN4-M87 were determined by molecular replacement using Phaser (*McCoy et al., 2007*). The search model was generated from the coiled-coil dimer structure of GCN4 fused to the coiled-coil dimer structure of striated muscle α-tropomyosin (PDB 1KQL) using Sculptor (*Bunkóczi and Read, 2011*) with default settings. Extensive model modification and building were performed with Coot and guided by the inspection of σA-weighted 2mFo-DFc and mFo-DFc omit maps (*Figure 2—figure supplement 1*).

LL-37 in the GCN4-M87/LL-37 complex was manually modeled into well-defined difference electron density that was visible after a few rounds of refinement of the search model. The asymmetric unit of the GCN4-M87/LL-37 crystal contained two heteromeric assemblies, each composed of two chains of GCN4-M87 and one of LL-37. Refinement was performed using Refine from the Phenix suite (*Afonine et al., 2012*) with default settings. At the final stages of refinement, TLS parameters were applied. TLS groups were applied as follows: chain A; chain B; chain C aa 41–55 and 56–93; chain D aa 38–55 56–104; chain E; and chain F. Side chains with no corresponding electron density were truncated to Cβ. Interaction interfaces between M87 and LL-37 were analyzed with PISA (*Krissinel and Henrick, 2007*).

The GCN4-M87 dimer was modeled into continuous electron density. The asymmetric unit of the GCN4-M87 crystal contained a single coiled-coil dimer. Refinement of GCN4-M87 was performed using Refine with default settings in addition to the application of twofold NCS restraints and TLS parameters. Each chain constituted a single TLS group. Three and six amino acids at N- and C-termini, respectively, of chain A and a single N-terminal amino acid of chain B lacked electron density and were not modeled.

Molecular figures were generated with PyMOL (http://pymol.sourceforge.net).

Structures have been deposited in the PDB: 7SAY for GCN4-M87/LL-37 and 7SAF for GCN4-M87.

## CD spectroscopy

CD spectra were measured on an Aviv 215 Circular Dichroism Spectrometer using a quartz cell with 1 mm path length. Protein samples were ~0.125–0.250 mg/ml in 5 mM sodium phosphate, pH 7.9. Wavelength spectra were recorded in a range of 190–260 nm at 37°C at 0.5 nm intervals with a 0.5 s averaging time per data point. Melting curves were determined at 222 nm between 20°C and 75°C with 1°C increments and 30 s equilibration time for each temperature point. Two independent experiments were carried out for each sample, and the data were averaged and presented as a mean molar residue ellipticity.

## Identification of M87 LL-37-binding motif in other M proteins

The sequence of the N-terminal 250 amino acids of mature M87 was aligned against the sequence of the N-terminal 250 amino acids of the mature form of 179 M proteins using Clustal Omega (*Sievers et al., 2011*). Alignments were manually curated for the presence of a Tyr (or Phe, although none were found) occupying the position equivalent to M87 Tyr81, the occurrence of hydrophobic amino

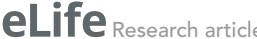

acids at the *d*, *a*, and *d* positions following the Tyr, and a hydrogen bond acceptor at the *e* position following the Tyr.

## Acknowledgements

We thank S Rees for help with crystallographic data collection. This work was supported by NIH 1R21AI144901 (PG). Use of the Stanford Synchrotron Radiation Lightsource, SLAC National Accelerator Laboratory, is supported by the U.S. Department of Energy, Office of Science, Office of Basic Energy Sciences under Contract No. DE-AC02-76SF00515. The SSRL Structural Molecular Biology Program is supported by the DOE Office of Biological and Environmental Research, and by the National Institutes of Health, National Institute of General Medical Sciences (P30GM133894).

## Additional information

### Funding

| Funder | Grant reference number | Author |
|---|---|---|
| National Institutes of Health | R21AI144901 | Partho Ghosh |

The funders had no role in study design, data collection and interpretation, or the decision to submit the work for publication.

### Author contributions

Piotr Kolesinski, Formal analysis, Investigation, Methodology, Validation, Visualization, Writing - original draft, Writing – review and editing; Kuei-Chen Wang, Formal analysis, Investigation, Methodology, Writing – review and editing; Yujiro Hirose, Resources, Writing – review and editing; Victor Nizet, Conceptualization, Funding acquisition, Writing – review and editing; Partho Ghosh, Conceptualization, Funding acquisition, Methodology, Project administration, Supervision, Visualization, Writing - original draft, Writing – review and editing

### Author ORCIDs

Piotr Kolesinski ⓘ http://orcid.org/0000-0001-6034-8445
Yujiro Hirose ⓘ http://orcid.org/0000-0001-6338-4767
Victor Nizet ⓘ http://orcid.org/0000-0003-3847-0422
Partho Ghosh ⓘ http://orcid.org/0000-0003-0254-5218

### Decision letter and Author response

Decision letter https://doi.org/10.7554/eLife.77989.sa1
Author response https://doi.org/10.7554/eLife.77989.sa2

## Additional files

### Supplementary files
• Transparent reporting form

• Source data 1. Source data for *Figure 1*, *Figure 1—figure supplement 1*, *Figure 4*, and *Figure 5*.

### Data availability
Structures have been deposited in the PDB: 7SAY for GCN4-M87/LL-37 and 7SAF for GCN4-M87.

The following datasets were generated:

| Author(s) | Year | Dataset title | Dataset URL | Database and Identifier |
|---|---|---|---|---|
| Kolesinski P, Wang K, Hirose Y, Nizet V, Ghosh P | 2022 | An M protein coiled coil unfurls and exposes its hydrophobic core to capture LL-37 | https://www.rcsb.org/structure/7SAY | RCSB Protein Data Bank, 7SAY |
| Kolesinski P, Wang K, Hirose Y, Nizet V, Ghosh P | 2022 | An M protein coiled coil unfurls and exposes its hydrophobic core to capture LL-37 | https://www.rcsb.org/structure/7SAF | RCSB Protein Data Bank, 7SAF |

The following previously published datasets were used:

| Author(s) | Year | Dataset title | Dataset URL | Database and Identifier |
|---|---|---|---|---|
| Li Y, Mui S, Brown JH, Strand J, Reshetnikova L, Tobacman LS, Cohen C | 2002 | Crystal structure of the C-terminal region of striated muscle alpha-tropomyosin at 2.7 angstrom resolution | https://www.rcsb.org/structure/1KQL | RCSB Protein Data Bank, 1KQL |
| Sancho-Vaello E, Gil-Carton D, Francois P, Bonetti EJ, Kreir M, Pothula KR, Kleinekathofer U, Zeth K | 2020 | The structure of the human tetrameric LL-37 peptide in a channel conformation | https://www.rcsb.org/structure/7PDC | RCSB Protein Data Bank, 7PDC |
| Sancho-Vaello E, Francois P, Bonetti EJ, Lilie H, Finger S, Gil-Ortiz F, Gil-Carton D, Zeth K | 2017 | The crystal structure of dimeric LL-37 | https://www.rcsb.org/structure/5NNM | RCSB Protein Data Bank, 5NNM |

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
