## [Editor Report]

In this exciting article, the authors solve the crystal structure of a complex of LL-37 with the M protein M87. In this structure, the M87 coiled coil unfurled to exposed its hydrophobic core to interact with LL-37. These studies have provided important new information regarding the mechanism of interaction between coiled-coil proteins and the α-helical LL-37.

---

## [Decision Letter]

**Decision letter after peer review:**

Thank you for submitting your article "An M protein coiled coil unfurls and exposes its hydrophobic core to capture LL-37" for consideration by *eLife*. Your article has been reviewed by 2 peer reviewers, one of whom is a member of our Board of Reviewing Editors, and the evaluation has been overseen by Volker Dötsch as the Senior Editor. The reviewers have opted to remain anonymous.

Essential revisions:

The reviewers were very positive about this, please see below for more details from each reviewer.

*Reviewer #1 (Recommendations for the authors):*

Line 76- I don't think the gels give information on binding capacity, this would require more quantitative assays to measure Kds. Although it looks like M87N100 pulled down more LL-37, it also looks like there is less M58N100 in the adjacent well, and there are two bands present there, making it impossible to compare these with each other. The Coomassie gels in Figure 1A should also be quantified across replicates.

Figure 1B needs stats, otherwise one cannot directly compare samples as done in Lines 83-85.

It's not clear why the authors used a truncated form of the M87 protein (68-105aa). In addition, what was the advantage of adding a portion of GCN4? What is this fragment? In general, more information in these regards is necessary for the more general audience, preferably before jumping into line 86.

It is not clear what 2C is showing, can the amino acids be marked on the image? The text indicates it should be a comparison of bound and unbound, but it looks to be in a different orientation?

Is there a way Figure 3 can illustrate more information about the types of contacts that are predicted to be occurring? The information is in the text, but it would be helpful to have them illustrated in the figure for reference.

In figure 4, given that the E85A mutation did not affect the interaction with LL-37, the interpretation would be that the electrostatic interaction with the Lysine in LL-37 is not important for the association of M87 with LL-37. The E85R mutant could be causing repulsion effects that are dominant negative. This should be considered, and it is not clear what is the basis for the conclusion in lines 167-168 "the polar contact conferred by M87 Glu85 conferred specificity rather than binding affinity". Data has yet to be presented to support this statement. Because of the caveat regarding the E85R mutant, it is important to perform the Strep experiment in Figure 4D with other M87 mutants with alanine substitutions. This also affects the data interpretations in Figure 5. Were alanine substitutions made here as well?

For figure 4B, how are stats determined if there is no variation in WT? Did the authors compare the WT values to the average WT to determine the spread and distribution of WT data? There are also error bars missing from most of the samples in 4B.

Line 70- the word "binding" (or something analogous) missing at beginning of the sentence: "The determine the mechanism of LL-37 by M proteins".

*Reviewer #2 (Recommendations for the authors):*

In terms of the general science and the way that the paper is written and presented, I can find little to fault. However, I think that the paper might be improved with a few additional considerations and small changes.

First of all, have Kds (dissociation constants) been measured for the LL-37 peptide binding to the M protein fragments? This would be useful quantitative information on the system anyway, but it could also inform on the mechanism and efficacy of action of the M proteins in sequestering the peptide.

Second, I agree that to my knowledge this unfurling (or what I would term strand-invasion) mechanism of binding is new in natural coiled-coil structures. However, there is one other recently described system that operates somewhat similarly. See: Dimitrova-Paternoga et al. "Molecular basis of mRNA transport by a kinesin-1-atypical tropomyosin complex", DOI: 10.1101/gad.348443.121; and PDB entry 7BJS. Whilst this is clearly very different in terms of proteins involve and the mechanism (the work that I point out is "strand-accretion" rather than "strand-invasion" per se), I think that it's worth discussing (briefly) and citing this paper in a revised manuscript.

Finally, I would encourage the authors to perform a SOCKET analysis of their structures (Kumar and Woolfson "Socket2: a program for locating, visualizing and analyzing coiled-coil interfaces in protein structures", DOI:10.1093/bioinformatics/btab631; http://coiledcoils.chm.bris.ac.uk/socket2/home.html). I did this quickly using the two structures provided by the authors using 7, 7.5, and 8 Å cutoffs, and I found it informative. Obviously, the GCN4 regions are positive for coiled-coil knobs-into-holes (KIH) interactions at all cutoffs. However, and in both structures, KIHs interactions in the M protein regions are actually quite sparse; though more are revealed as the threshold is increased-with only I84 apparent at 7 Å, then 84 +L88 at 7.5 Å, and these plus Y81 at 8 Å. First, this does mean that the authors should modify their description of that region a little as it does not have many a + d positions engaged in KIH interaction. Moreover, and more interestingly, this suggests to me at least that the M protein coiled coils are not-ideal, or weak, and therefore potentially receptive to unfurling and strand invasion. This would be a neat bit of coiled-coil biology and evolution if it proved to be the case.

---

## [Author Response]

Reviewer #1 (Recommendations for the authors):Line 76- I don't think the gels give information on binding capacity, this would require more quantitative assays to measure Kds. Although it looks like M87N100 pulled down more LL-37, it also looks like there is less M58N100 in the adjacent well, and there are two bands present there, making it impossible to compare these with each other. The Coomassie gels in Figure 1A should also be quantified across replicates.

The reviewer makes a good point. We have revised the text accordingly to remove comparisons between M proteins (l. 75):

“M58^N100^ and M87^N100^ bound LL-37, as did the M1^HB^ fragment (Figure 1A), while the others did not bind LL-37 above background level (Figure 1 — figure supplement 1).*”*

Figure 1B needs stats, otherwise one cannot directly compare samples as done in Lines 83-85.

We have added statistics to the plot. In addition, since MIC’s are discrete quantities, we have revised the presentation of these data from bars to points (Figures 1B and 4D).

It's not clear why the authors used a truncated form of the M87 protein (68-105aa). In addition, what was the advantage of adding a portion of GCN4? What is this fragment? In general, more information in these regards is necessary for the more general audience, preferably before jumping into line 86.

We have added the following to clarify this point (l. 87):

“Various fragments of M58 and M87 were tried in co-crystallization trials with LL-37. Success was had with a version of M87 protein (amino acids, aa 68-105) that had a portion of the canonical coiled-coil protein GCN4 (aa 250-278) (8, 15, 18) fused in register to its N-terminus. GCN4 fusion was pursued to stabilize coiled-coil formation in the M87 protein fragment, a technique that has proven successful in the crystallization of a number of coiled-coil proteins (19, 20).”

It is not clear what 2C is showing, can the amino acids be marked on the image? The text indicates it should be a comparison of bound and unbound, but it looks to be in a different orientation?

It is indeed a comparison of bound and unbound M87 protein. We have marked the amino acid numbers on Figure 2C.

Is there a way Figure 3 can illustrate more information about the types of contacts that are predicted to be occurring? The information is in the text, but it would be helpful to have them illustrated in the figure for reference.

We added labels to the figure itself.

In figure 4, given that the E85A mutation did not affect the interaction with LL-37, the interpretation would be that the electrostatic interaction with the Lysine in LL-37 is not important for the association of M87 with LL-37. The E85R mutant could be causing repulsion effects that are dominant negative. This should be considered, and it is not clear what is the basis for the conclusion in lines 167-168 "the polar contact conferred by M87 Glu85 conferred specificity rather than binding affinity". Data has yet to be presented to support this statement. Because of the caveat regarding the E85R mutant, it is important to perform the Strep experiment in Figure 4D with other M87 mutants with alanine substitutions. This also affects the data interpretations in Figure 5. Were alanine substitutions made here as well?

Our interpretation is supported by the classic results of Clackson and Wells (*Science* 1995) on binding hot spots in protein-protein interactions, and much subsequent work since this seminal paper. This body of work has shown that protein-protein affinity mainly depends on hydrophobic interactions (entropic contributions due to the release of waters into the bulk environment in the bound state that are otherwise organized around hydrophobic groups in the free state), whereas polar interactions contribute little to affinity (as a protein-protein electrostatic interactions in the bound state are replaced by electrostatic protein-water interactions in the free state, each having similar energies and therefore providing no substantial net contribution to affinity). However, polar interactions do confer specificity based on charge complementarity (e.g., negative group on one protein adjacent to a positive group on the second). E85 has the characteristics of an amino acid that provides specificity rather than affinity — i.e., no difference in affinity when the side chain is removed (i.e., Ala substitution) but abrogation of binding when the character of the side chain is reversed (i.e., Arg substitution). We have modified the text to make this point more clearly (l. 163):

“Ala-substitution of M87 Glu85 had little effect on LL-37 binding (Figures 4A and B), suggesting it did not contribute to affinity. To ask whether it instead contributed to specificity, we asked whether Arg-substitution of this amino acid would decrease interaction with LL-37, as M87 Glu85 was adjacent to LL-37 Lys12.”

For figure 4B, how are stats determined if there is no variation in WT? Did the authors compare the WT values to the average WT to determine the spread and distribution of WT data? There are also error bars missing from most of the samples in 4B.

We thank the reviewer for bringing this to our attention. We applied the normalization at the incorrect stage in our initial submission, resulting in all WT values being 1.0 and with no standard deviations. We have corrected that, and there are now standard deviations in Figure 4B. We have also added a description in Materials and methods as to how the data were handled (l. 364):

“The intensity of gel bands was quantified with ImageJ (38) and confirmed to be within the linear range of detection. As the intensity on each gel depended on the length of time of staining and destaining, a normalization factor was used. For this, the ratio of the LL-37 band intensity to wild-type M protein band intensity from a particular gel lane was determined from three or more independent gels, and the mean of these ratios (mean of LL-37/M_WT_) was used as the normalization factor. Accordingly, for each data point (i.e., gel lane), the ratio of the LL-37 band intensity to M protein (wild-type or mutant) band intensity was quantified, and divided by the normalization factor (mean of LL-37/M_WT_).”

And also in the Figure Legends (l. 673):

“B. Quantification of LL-37 binding to intact wild-type and mutant M87 proteins. The ratio between LL-37 and wild-type M87 protein band intensities was determined in four independent experiments, and used to determine a mean LL-37/wt-M87 ratio. Points shown are ratios between LL-37 and (wild-type or mutant) M87 protein band intensities, normalized by the mean LL-37/wt-M87 ratio. For each sample, means and SD are shown. Statistical significance was calculated using one-way ANOVA with Dunnett post-hoc test. P values are as follows: ns > 0.05, * ≤ 0.05, ** ≤ 0.01, *** ≤ 0.001 and **** ≤ 0.0001.”

Line 70- the word "binding" (or something analogous) missing at beginning of the sentence: "The determine the mechanism of LL-37 by M proteins".

Thank you for catching this. We have inserted “binding.”

Reviewer #2 (Recommendations for the authors):In terms of the general science and the way that the paper is written and presented, I can find little to fault. However, I think that the paper might be improved with a few additional considerations and small changes.First of all, have Kds (dissociation constants) been measured for the LL-37 peptide binding to the M protein fragments? This would be useful quantitative information on the system anyway, but it could also inform on the mechanism and efficacy of action of the M proteins in sequestering the peptide.

The K_D_ between M1 protein and LL-37 has been measured to be ~1 μM, and we have added that information to the paper (l. 77): “As a point of comparison, the K_D_ between M1 protein and LL-37 is ~1 μM (9).” We do not know whether the LL-37 binding mode for M1 protein is as complex as we for M87 protein, with sequential addition of LL-37 molecules. However, it appears by the semi-quantitative method of co-precipitation, the affinities of M58 and M87 for LL-37 are similar to that of M1 protein.

Second, I agree that to my knowledge this unfurling (or what I would term strand-invasion) mechanism of binding is new in natural coiled-coil structures. However, there is one other recently described system that operates somewhat similarly. See: Dimitrova-Paternoga et al. "Molecular basis of mRNA transport by a kinesin-1-atypical tropomyosin complex", DOI: 10.1101/gad.348443.121; and PDB entry 7BJS. Whilst this is clearly very different in terms of proteins involve and the mechanism (the work that I point out is "strand-accretion" rather than "strand-invasion" per se), I think that it's worth discussing (briefly) and citing this paper in a revised manuscript.

We thank the reviewer for bringing this paper to our attention, and we have added the following to the Discussion (l. 256):

“Recently, a coiled-coil dimer that binds an α-helical partner in a roughly similar manner was reported (29), but in that complex coiled-coil character is maintained, unlike the loss of coiled-coil character observed for the M87/LL-37 complex.”

Finally, I would encourage the authors to perform a SOCKET analysis of their structures (Kumar and Woolfson "Socket2: a program for locating, visualizing and analyzing coiled-coil interfaces in protein structures", DOI:10.1093/bioinformatics/btab631; http://coiledcoils.chm.bris.ac.uk/socket2/home.html). I did this quickly using the two structures provided by the authors using 7, 7.5, and 8 Å cutoffs, and I found it informative. Obviously, the GCN4 regions are positive for coiled-coil knobs-into-holes (KIH) interactions at all cutoffs. However, and in both structures, KIHs interactions in the M protein regions are actually quite sparse; though more are revealed as the threshold is increased-with only I84 apparent at 7 Å, then 84 +L88 at 7.5 Å, and these plus Y81 at 8 Å. First, this does mean that the authors should modify their description of that region a little as it does not have many a + d positions engaged in KIH interaction. Moreover, and more interestingly, this suggests to me at least that the M protein coiled coils are not-ideal, or weak, and therefore potentially receptive to unfurling and strand invasion. This would be a neat bit of coiled-coil biology and evolution if it proved to be the case.

We thank the reviewer for bringing this to our attention, and we have added this point to the Discussion (l. 250):

“Intriguingly, analysis of ‘knobs-into-holes’ packing of M87 in the free state shows that A77 and Y81 are not ideally packed (28). This suggests that the segment containing A77 and Y81 may initiate unfurling to bind LL-37.”

And (l. 280):

“The loose ‘knobs-into-holes’ packing of Y81 in the free form of M87 along with the conservation of the Tyr in M proteins shown or implicated to bind LL-37 suggests that this may be a general site for initiating unfurling.”